# *VTRNA2-1*: Genetic Variation, Heritable Methylation and Disease Association

**DOI:** 10.3390/ijms22052535

**Published:** 2021-03-03

**Authors:** Pierre-Antoine Dugué, Chenglong Yu, Timothy McKay, Ee Ming Wong, Jihoon Eric Joo, Helen Tsimiklis, Fleur Hammet, Maryam Mahmoodi, Derrick Theys, John L. Hopper, Graham G. Giles, Roger L. Milne, Jason A. Steen, James G. Dowty, Tu Nguyen-Dumont, Melissa C. Southey

**Affiliations:** 1Precision Medicine, School of Clinical Sciences at Monash Health, Monash University, Clayton, VIC 3168, Australia; Pierre-Antoine.Dugue@monash.edu (P.-A.D.); chenglong.yu@monash.edu (C.Y.); tmck0012@student.monash.edu (T.M.); eeming.wong@monash.edu (E.M.W.); helen.tsimiklis@monash.edu (H.T.); fleur.hammet@monash.edu (F.H.); maryam.mahmoodi@monash.edu (M.M.); derrick.theys@monash.edu (D.T.); Graham.Giles@cancervic.org.au (G.G.G.); Roger.Milne@cancervic.org.au (R.L.M.); jason.steen@monash.edu (J.A.S.); tu.nguyen-dumont@monash.edu (T.N.-D.); 2Cancer Epidemiology Division, Cancer Council Victoria, Melbourne, VIC 3004, Australia; 3Centre for Epidemiology and Biostatistics, Melbourne School of Population and Global Health, The University of Melbourne, Parkville, VIC 3010, Australia; j.hopper@unimelb.edu.au (J.L.H.); jdowty@unimelb.edu.au (J.G.D.); 4Department of Clinical Pathology, Melbourne Medical School, The University of Melbourne, Parkville, VIC 3010, Australia; ji.joo@unimelb.edu.au; 5Kathleen Cuningham Foundation Consortium for Research into Familial Breast Cancer, Research Division, Peter MacCallum Cancer Centre, Melbourne, VIC 3000, Australia; heather.thorne@petermac.org

**Keywords:** *VTRNA2-1*, nc886, MIR886, methylation quantitative trait loci, SNP-based heritability, rs2346018, breast cancer, prostate cancer

## Abstract

*VTRNA2-1* is a metastable epiallele with accumulating evidence that methylation at this region is heritable, modifiable and associated with disease including risk and progression of cancer. This study investigated the influence of genetic variation and other factors such as age and adult lifestyle on blood DNA methylation in this region. We first sequenced the *VTRNA2-1* gene region in multiple-case breast cancer families in which *VTRNA2-1* methylation was identified as heritable and associated with breast cancer risk. Methylation quantitative trait loci (mQTL) were investigated using a prospective cohort study (4500 participants with genotyping and methylation data). The *cis*-mQTL analysis (334 variants ± 50 kb of the most heritable CpG site) identified 43 variants associated with *VTRNA2-1* methylation (*p* < 1.5 × 10^−4^); however, these explained little of the methylation variation (R^2^ < 0.5% for each of these variants). No genetic variants elsewhere in the genome were found to strongly influence *VTRNA2-1* methylation. SNP-based heritability estimates were consistent with the mQTL findings (h^2^ = 0, 95%CI: −0.14 to 0.14). We found no evidence that age, sex, country of birth, smoking, body mass index, alcohol consumption or diet influenced blood DNA methylation at *VTRNA2-1*. Genetic factors and adult lifestyle play a minimal role in explaining methylation variability at the heritable *VTRNA2-1* cluster.

## 1. Introduction

Mendelian-like inheritance of germline DNA methylation can be due to *cis*- or *trans*-acting genetic factors known as methylation Quantitative Trait Loci (mQTL) or epimutations (heritable change in gene activity that is not associated with a DNA mutation but rather with gain or loss of DNA methylation or other heritable modification of chromatin). Both can mimic germline pathogenic variants in their effect on gene function and disease association and discriminating between the two possibilities (mQTL or epimutation) in specific genomic regions and disease context is often challenging.

We previously made a genome-wide assessment of heritable methylation using a family design [1]; probes were ranked by a methylation-heritability metric and 24 of the 1000 most heritable CpGs were identified to be associated with breast cancer risk in these families. Several CpGs within *VTRNA2-1* were among those that appeared to be most heritable and most strongly associated with breast cancer risk, including five within the gene promoter [1]. Evidence that DNA methylation can be transmitted from parent to offspring in the absence of a genetic explanation is scarce and controversial [2,3,4]. It is therefore important to assess whether any genetic variation may influence DNA methylation at the *VTRNA2-1* region, i.e., *cis-* or *trans-*acting mQTL.

*VTRNA2-1* has been demonstrated to adopt two structurally and functionally distinct RNA conformations, one which strongly inhibits protein kinase R (PKR) and downstream eukaryotic translation initiation factor 2 subunit α (eIF2α) phosphorylation, and one which acts as a pseudo-inhibitor of PKR when competing with other double-stranded (ds) RNA molecules [5]. The ~2 kb region overlapping the *VTRNA2-1* locus has been reported to be polymorphic or atypically imprinted, and somatically acquires DNA methylation of the maternal allele in the majority of cases [6,7]. PKR is an interferon-induced kinase consisting of 551 amino acids that acts as an intracellular stress sensor, primarily associated with viral infection. dsRNA produced by viral replication binds to and activates PKR, causing dimerisation and subsequent phosphorylation of its substrate, eIF2α. Phosphorylation of eIF2α converts eIF2 from a substrate to an inhibitor of its GDP-GTP exchange factor eIF2B, inhibiting mRNA translation and arresting global protein synthesis [8,9]. In addition to eIF2α phosphorylation, PKR can activate the nuclear factor kappa-light-chain-enhancer of the activated B cells (NF-κB) signalling pathway, which is known to play an oncogenic role in tumourigenesis [10]. PKR is reported to act on the NF-κB pathway by inducing phosphorylation of IκBα, which interestingly requires the expression of *VTRNA2-1* to occur [9].

There is evidence that *VTRNA2-1* may act as a tumour suppressor and is a metastable epiallele [11,12]. The study by van Baak et al. used data from the Melbourne Collaborative Cohort Study (MCCS) to assess the association between methylation at metastable epialleles and cancer risk and concluded that methylation at *VTRNA2-1* was potentially associated with risk of lung cancer and B-cell lymphoma [13]. We also showed that *VTRNA2-1* promoter methylation was associated with prostate cancer risk, and these associations appeared stronger for aggressive disease [14].

Consistent with a familial aggregation of *VTRNA2-1* methylation not being due to genetic factors are observations that DNA methylation at this locus is sensitive to the perinatal environment, including factors such as season of conception [13,15] or maternal folic acid supplementation during pregnancy [16]. That *VTRNA2-1* methylation at birth was found to be associated with childhood overweight/obesity [17] may signal another plausible link with cancer risk in adulthood, in addition to the tumour suppressor role [15].

DNA methylation at *VTRNA2-1* has therefore been hypothesised to show the following pattern [7]: (1) The paternally-inherited allele seems to always be unmethylated, as observed in several studies [1,14], and (2) the maternally-inherited allele is methylated in ~75% and unmethylated in ~25% of individuals. Methylation at this locus is thought to be influenced by the aforementioned pre-/perinatal environmental factors, as well as by genetic variants, the latter possibly due the role of CTCF (transcription factor, CCCTC-binding) in imprinting via the influence of rs2346018. A recently published study of genome-wide mQTLs in 27,750 European participants [18] revealed relatively weak associations between genetic variants and methylation at cg26328633 (CpG site identified as part of the strong heritable cluster in our Australian families), but no data were available for neighbouring CpGs, and no apparent association was found for other single nucleotide polymorphisms (SNP) of interest within this region, including rs2346018. No mQTLs were found for any of the most heritable *VTRNA2-1* DNA methylation marks in a previous genome-wide assessment that included ~2000 participants [19].

In this study, our aims were three-fold: first, to sequence the *VTRNA2-1* region to assess the presence of rare genetic variation at this locus; second, to conduct a genome-wide assessment of mQTLs and SNP-based methylation heritability in the *VTRNA2-1* region (previously identified heritable marks); third, to assess whether any genetic variants associated with DNA methylation in this region contribute to the previously observed associations with breast cancer risk.

## 2. Results

This study used data from (1) 179 participants of multiple-case breast cancer family studies to assess the presence of genetic variants in the *VTRNA2-1* heritable region, (2) 4500 participants in a prospective study to assess mQTLs and SNP-based heritability (genome-wide and focusing on *cis-*variants), and (3) 2141 participants in breast cancer family-based studies to adjust the *VTRNA2-1* results of our previous publication [1] for a nearby SNP.

The characteristics of participants in the prospective cohort (Melbourne Collaborative Cohort Study) and multiple-case family (Australian Breast Cancer Family Registry and Kathleen Cuningham Foundation Consortium for research into Familial Breast Cancer) studies (see Methods Section) included in this study are shown in Table 1. In the prospective cohort study, the majority were male, aged between 50 and 70 and never or former smokers. The distribution of methylation beta values at the *VTRNA2-1* region is shown in Figure 1b,c. Nine genetic variants were identified via sequencing (Figure 1a) in members of multiple-case breast cancer families. Of these, one was rare (identified in only one participant) and was excluded from further analysis. The other eight variants were used to estimate carrier probabilities in members of multiple-case breast cancer families (see Methods Section; *4.4.4. Associations with breast cancer risk*). All eight variants were available after genotype imputation in the prospective cohort study (see Methods Section; *4.2. Genetic and methylation data*) and therefore included in the mQTL analysis. None of these variants were found to directly overlap with the most heritable *VTRNA2-1* methylation site (cg06536614).

Methylation values at the five most heritable methylation marks (cg06536614, cg26328633, cg25340688, cg26896946 and cg00124993) were highly correlated (in the MCCS all r ≥ 0.88); we therefore focused on cg06536614, which was found to be the most heritable methylation mark [1]. The percentage methylation at cg06536614 was lower than 50% for 1946 (43%) of participants and lower than 60% for 4414 (98%) of them; 23% had less than 30% methylation (Figure 2).

The mQTL analysis was performed in 4500 participants in the prospective, population-based study (MCCS) for a total of 10,484,498 genetic variants; a *cis-*mQTL analysis was then carried out by focusing on genetic variants within 50 kb of the most heritable methylation mark (cg06536614). Genome-wide, we found no evidence that any included genetic variant was associated with *VTRNA2-1* methylation (all *p* > 5 × 10^−9^), with similar results obtained for the M-value (logit transformation of beta value) or RINT (rank-based inverse normal transformation, which was applied previously in the context of DNA methylation analyses and provides a Gaussian methylation distribution, which is not always the case for the M-values) transformation (Appendix A, respectively, showing the 100 CpGs with smallest *p*-values). Results from the *cis*-mQTL analysis, i.e., genetic variants within 50 kb of the CpG, are shown in Appendix A (M-values) and Appendix A (RINT-values). There were 43 variants with *p* < 1.5 × 10^−4^, indicating evidence of genetic influences on *VTRNA2-1* methylation (Table 2 and Figure 3). The strongest evidence of association was observed for rs2190622 (*p* = 5 × 10^−6^) (Figure 2), although this association was not substantially stronger than for other neighbouring SNPs. A significant association was also observed for rs2346018 (*p* = 8 × 10^−5^) (Figure 2), which was previously reported to modify methylation in this region [7]. Consistent with this, associations appeared stronger for the variants located closer to the CpG of interest (Appendix A and Figure 3). Associations were qualitatively similar but appeared somewhat stronger for the RINT-transformed values (strongest hit: rs2190622, *p* = 8 × 10^−8^; rs2346018: *p* = 8 × 10^−6^ and Appendix A and Figure 3). These variants appeared to explain very little of variation in *VTRNA2-1* methylation (Figure 2) with a variance explained ranging from 0.33% to 0.47%. The findings were similar when restricting the analyses to MCCS participants selected as controls (not shown).

Results from SNP-based heritability analyses (i.e., taking into account > 1M variants) are shown in Table 3. These were consistent with the mQTL analyses in showing minimal influence of SNPs on methylation in the region: h^2^ = 0, 95%CI: −0.14 to 0.14. These results were virtually the same when the RINT transformation was used instead of M-values (Table 3) or restricting the analyses to MCCS participants selected as controls (not shown). Non-null or high heritability was observed for methylation sites distant from the heritable *VTRNA2-1* region (>2–8 kb, Table 3).

The associations between DNA methylation M-values and participant characteristics age, sex, country of birth and lifestyle factors (tobacco smoking, body mass index, alcohol consumption and healthy eating) are shown in Table 4. None of these factors showed an association with *VTRNA2-1* methylation; results were similar when considering only MCCS participants who were selected as controls or using RINT-values instead of M-values (results not shown). There was also no apparent influence of the sample proportion of white blood cells on DNA methylation (Table 4).

Associations of *VTRNA2-1* methylation (cg06536614) with breast cancer risk were assessed in 2141 participants in multiple-case breast cancer families, using the same methods as in our previous publication [1]. As genotypes were not directly measured in all family members, we estimated carrier probabilities for genetic variants, using a method similar to that used to estimate methylation carrier probabilities [1], based on sequencing data for 179 participants and imputed OncoArray data for 23 participants (all multiple-case family members); the association of methylation with breast cancer risk was therefore assessed using the same models as previously, with additional adjustment for genetic variant carrier probabilities. The results “Not adjusted for SNPs” are the same as those presented in Joo et al. [1]. The association of cg06536614 methylation with breast cancer risk remained highly significant after adjustment for rs2346018 carrier probabilities, with *p* values ranging from 2 × 10^−9^ to 3 × 10^−8^ in the unadjusted model and from 8 × 10^−9^ to 1 × 10^−7^ in the adjusted model (Table 5). Similar results were obtained after adjustment for any of the other three variants significantly associated with cg06536614 (Table 2) for which carrier probabilities could be estimated (data not shown).

## 3. Discussion

Our study provides further evidence that DNA methylation at *VTRNA2-1* is minimally influenced by genetic factors, and thus, the Mendelian-like inheritance of germline DNA methylation at this locus is likely to be via a true epimutation mechanism rather than via a mQTL. Therefore, the “missing heritability” (approximately the difference between family-based and SNP-based heritability) appears to be substantial, which confirms the findings of Joo et al. [1]. Genetic variants are therefore unlikely to fully explain any associations between methylation at this locus and disease risk, including breast cancer as we found previously in the context of a multiple-case breast cancer family study.

Some genetic variants at this locus were statistically significantly associated with blood DNA methylation (carriers of the minor allele being less likely to show hypomethylation), but the effect sizes and variance explained were small (variance explained 0.4% to 0.5% for the strongest individual associations). Rs2346018 [6,7], which was previously implicated with methylation at this locus, showed a significant association with methylation in our data, thereby confirming it may exert a small influence. Although our study does not allow disentangling which of the identified SNPs might causally affect DNA methylation, it should be noted that (1) rs2346018 was one of the SNPs with smallest *p*-value among the 334 tested, and (2) many SNPs that appeared most strongly associated with methylation did not have a clear functional interpretation, for example rs2190622, the strongest observed association, is located in an intergenic region and has unknown regulatory function (Appendix A).

In their genome-wide assessment of imprinting in the methylome, Zink et al. concluded that *VTRNA2-1* is an example of a region with polymorphic imprinted methylation unrelated to SNP genotypes. In our study, a large number of participants (45%) had a percentage of methylation between 50% and 60%. Although this might appear to be inconsistent with maternal imprinting, this could indicate the limitation of the HM450 assay to measure DNA methylation with sufficient precision or DNA methylation at the paternal allele accumulated over the lifetime. However, technical validation using PyroMark (Pyrosequencing technical validation) produced similar methylation values, and we did not find evidence that *VTRNA2-1* methylation was influenced by factors for which our data shows widespread methylation changes, such as age, sex, country of birth, or other factors shown to strongly affect DNA methylation such as tobacco smoking, alcohol consumption or body mass index [20,21,22]. In this study, we did not have information on early-life factors. Although several factors in utero and early in life were shown to modulate DNA methylation at *VTRNA2-1*, none of the findings presented for season of conception [15], maternal folate during pregnancy [16], or gestational famine exposure [23] appear to fully explain methylation variation in this region; the mechanisms by which *VTRNA2-1* methylation is inherited therefore appear to be essentially non-genetic and only partially explained by the factors studied in the literature so far.

Previous studies of mQTLs and heritability have only provided a partial assessment of genetic influences on *VTRNA2-1* methylation. The study by Gaunt et al. [24] also did not report SNP-based heritability but assessed mQTLs. Various potential trans-mQTLs were identified but evidence of associations appeared weak (all *p* > 3 × 10^−8^, http://www.mqtldb.org/search.htm, accessed on 1 December 2020). The large meta-analysis by Min et al. only reported significant associations for cg26328633 based on 27,750 European participants; 12 SNPs located close to *VTRNA2-1* had a *p*-value ranging from 10^−22^ to 10^−26^ (http://mqtldb.godmc.org.uk/, accessed on 1 December 2020), but the effect sizes were virtually the same as those obtained in our study (beta ~0.1) so that the variance explained was likely similarly small. Results for rs2346018 or other variants of interest were not provided. In the study by McRae et al. [25], the family-based heritability (peripheral blood leukocytes) of our five most heritable marks associated with breast cancer was ~0.50. The study by van Dongen et al. [26] found very high twin-based heritability (~0.97) for the five *VTRNA2-1* CpGs but SNP-based heritability was reported as “NA” (due to convergence problems using GCTA, which might mean those values were in fact equal to zero). The *p* value cut-off we used for detecting mQTLs genome-wide was conservative (strict Bonferroni correction, *p* = 5 × 10^−9^), but using other cut-offs commonly used in genome-wide association studies, such as *p* = 5 × 10^−8^, would have resulted in the same conclusion (M-values: one intergenic variant in chromosome 1, *p* = 3 × 10^−8^ and explaining 0.7% of methylation variability; RINT values: no variant with *p* < 5 × 10^−8^). It is possible that some true associations were not detected, but these would likely have a weak influence on DNA methylation. Other studies used various cut-offs to declare statistically significant mQTLs, e.g., *p* = 10^−14^ in Gaunt et al. [24], *p* = 10^−11^ in McRae et al. [19], and *p* = 10^−8^ in Min et al. [18], but these studies carried out more tests because they investigated >400,000 CpGs.

Although findings are difficult to compare across studies, they appear to be consistent with ours in showing weak influences of genetic variants on *VTRNA2-1* methylation. Consistent with this, the associations of DNA methylation with breast cancer risk after adjustment for rs2346018 were only slightly attenuated. It should also be noted that none of rs2346018 or other sequenced genetic variants were found to be associated with breast cancer risk in the largest genome-wide association studies to date [27,28].

Although all methylation measures in this study were made on blood samples, it is worth noting that *VTRNA2-1* methylation has been implicated as playing a role in, or being influenced during, carcinogenesis. Fort et al. [29] sought to find direct association between *VTRNA2-1* transcript levels and methylation of its promoter in prostate tumour samples. Average *VTRNA2-1* promoter methylation was found to be substantially increased in both low-grade and metastatic tumour tissue compared with normal prostate tissue. Additionally, average *VTRNA2-1* promoter methylation appeared to correlate with Gleason score, clinical T-value and biochemical relapse [29]. The levels of *VTRNA2-1* transcript were found to be inversely correlated with average promoter methylation. The relationship between *VTRNA2-1* and cancer growth appears to be tissue specific, with several studies suggesting a tumour suppressive role, e.g., cholangiocarcinoma [30], oesophageal carcinoma [31], small cell lung cancer [32], gastric cancer [33] and acute myeloid leukaemia [34], and some suggesting an oncogenic role, e.g., in endometrial cancer [35] and thyroid cancer [36].

We conclude that the genetic and non-genetic factors we investigated play a minimal role in explaining variation in blood DNA methylation at *VTRNA2-1*, so these are unlikely to play a strong role in observed associations between *VTRNA2-1* methylation and disease risk. The mechanism of inheritance of DNA methylation in this region remains to be elucidated.

## 4. Materials and Methods

### 4.1. Data Sources

#### 4.1.1. Prospective Cohort Study

The Melbourne Collaborative Cohort Study (MCCS) is a community-based study that recruited 41,513 participants in 1990–1994 [37]. Several nested case-control studies were conducted to assess associations between DNA methylation in blood and the risk of eight types of cancer. Incident cases were matched to controls on age, sex, country of birth and sample type (buffy coats/dried blood spots/peripheral blood mononuclear cells) using incidence density sampling [37,38]. We also used questionnaire-collected data on smoking and alcohol consumption [20,21,22], measures of body mass index and derived a healthy eating index using a validated 121-item food frequency questionnaire [39,40].

#### 4.1.2. Multiple-Case Breast Cancer Families

A total of 210 individuals from 25 multi-generational multiple-case breast cancer families, including 20 from Kathleen Cuningham Foundation Consortium for research into Familial Breast Cancer (kConFab) and 5 from the Australian Breast Cancer Family Registry (ABCFR)), were included in this study [41,42,43]. Among these family members, there were 87 breast cancer cases and 123 unaffected relatives.

### 4.2. Genetic and DNA Methylation Data

The *VTRNA2-1* region (GRCh37, ch5:g.135414615-135417597) was screened in 179 of 210 individuals from a multiple-case breast cancer family by targeted-sequencing using a custom-designed HaloPlexHS panel (Agilent, Santa Clara, CA, USA). Libraries were prepared from blood-derived DNA according to the manufacturer’s instructions and sequenced on a 2 × 150 bp high-output flow cell on the HiSeq3000 (Illumina, San Diego, CA, USA). Paired-end reads were aligned to the human reference genome GRCh37 using BWA-mem 0.7.17 [44]. Adapter sequences were removed, and unique molecular indices were marked for downstream read-deduplication using the Agilent Genomics NextGen Toolkit (Agilent, Santa Clara, CA, USA). Target coverage was calculated using bedtools v.2.27.1 [45], and variant calling was performed using VarDict v.1.7 [46]. Genetic variants filtering was performed as described previously [47]. Variant annotation was performed on variants with a read depth ≥30× and a variant allele frequency (VAF) ≥0.15, using VarSeq VSClinical v2.2 (Golden Helix Inc., Bozeman, MT, USA). Finally, an additional 23 participants in ABCFR/kConFab had genetic measures made using OncoArray, using the same method as for the MCCS for genotype imputation (see next paragraph).

Genome-wide genotyping was conducted on blood DNA samples from 12,584 MCCS participants using the Infinium OncoArray-500K BeadChip (Illumina, San Diego, CA, USA) [37,48]. Following previous standardised protocols [28], we imputed autosomal genotypes using the Michigan imputation server [49] and IMPUTE version 2 [50] with the 1000 Genomes Project dataset (phase 3) as the reference panel. The genotype probabilities from imputation were used to hard-call (uncertainty < 0.1) the genotypes for variants with an imputation info score  >  0.3. For the current analysis, we included 4748 participants for whom DNA methylation data was also available. We then retained the hard-called variants with minor allele frequency > 0.001, missing genotype rate < 0.2 and Hardy–Weinberg equilibrium *p*-value  >  10^−6^. Furthermore, to avoid bias due to confounding by shared environment among close relatives, participants were removed based on relatedness, i.e., excluding one participant randomly selected from any pair with a genetic relationship ≥ 0.05 (4th-degree or closer relationship) [51,52]. This procedure also removed duplicated methylation samples (genetic relationship = 1) [38]. After these quality control steps, 4500 paired genetic-methylation samples were retained (including 2228 cancer cases and 2272 controls) and 10,484,498 genetic variants (including 9,551,474 SNPs) for the analysis.

For all samples, DNA methylation was measured using the HumanMethylation450 (HM450) BeadChip (Illumina, San Diego, CA, USA) using methods described previously [1,38,53]. We used methylation M-values as their distribution is usually closer to Gaussian than methylation beta-values [54]. As a sensitivity analysis, we performed a more direct normalization of beta-values using rank-based inverse normal transformation (RINT) which was applied previously in the context of DNA methylation analyses and provides a Gaussian methylation distribution, which is not the case for the M-values [55,56].

### 4.3. Technical Validation of Methylation Measures Using Pyrosequencing

Pyrosequencing (PSQ) conducted on the PyroMark Q48 (Qiagen, Hilden, Germany) was used to validate the methylation measures made on the HM450 assay. DNA was extracted as described previously [1] and bisulfite converted using Zymo (Zymo Research, Irvine, CA, USA). Forward, reverse and sequencing PCR primers were designed using PyroMark Assay Design Software version 2.0. (Forward primer sequence: 5Bios 5′-G/GGAGGAATTGAGAGTTTTTTTAGGATA-3′; Reverse primer sequence: 5′-CCTTCAAAATAACACCAACTTATATTATCA-3′; Sequencing primer sequence: 5′-ACATAAAAAAATCAATAAACACC-3′) to target cg04481923 and were synthesised by Integrated DNA Technologies (Coralville, IA, USA). The EpiTect PCR control DNA set (Qiagen, Hilden, Germany), which includes a completely methylated and a completely unmethylated bisulfite-converted control DNAs, was used to generate a standard curve. The EpiTect control DNAs were mixed in known ratios (0, 0.25, 0.50, 0.75 and 1) and run with ten test samples, along with a non-converted DNA sample and a no-template control. The PyroMark PCR cycling protocol was as follows: denaturation for 15 min at 95 °C, then 45 cycles of 30 s at 94 °C, 30 s at 56 C, and 30 s at 72 °C, then final extension for 10 min at 72 °C. For each sample, the raw percentage of methylation was determined at cg04481923 (*VTRNA2-1*) and calibrated using the EpiTect control DNAs standard curve.

### 4.4. Statistical Analysis

#### 4.4.1. Assessment of mQTLs

After QC of methylation and genetic data, 4500 participants (including 2228 cancer cases and 2272 controls) and 10,484,498 genetic variants (including 9551,474 SNPs) were available for the analysis. We first removed factors that may confound DNA methylation values using linear mixed models with methylation M-values (or RINT-values) as the outcome and as covariates: age, sex, sample type, white blood cell proportions (estimated using the Houseman algorithm [57,58]) and 20 genetic principal components to account for population structure/ancestry as fixed effects; and as random effects: study, plate and slide of the assay. Our sample therefore included both cancer cases and controls. The inclusion of cancer cases may bias mQTL associations because of collider bias [59]; collider bias is usually considered to be small [60,61], and this may be particularly true in our setting because no strong associations of individual methylation markers with cancer risk were observed, but we nevertheless assessed consistency of associations in controls by analysing them separately.

As slightly over 10M genetic variants were tested, we used the Bonferroni correction for multiple testing and considered associations with a *p*-value less than 0.05/10^−7^ = 5 × 10^−9^ to be potential true signals. Further, because *cis* acting genetic variants are considerably more likely than *trans* acting variants to influence DNA methylation, we considered all SNPs within 50 kb pairs of the methylation sites analysed. A total of 334 variants were identified, so we corrected the *cis*-mQTL analyses for multiple testing using the Bonferroni cut-off *p* = 1.5 × 10^−4^ (0.05/334).

#### 4.4.2. SNP-Based Heritability

The univariate genome-based restricted maximum likelihood (GREML) method [62,63] was used to estimate the SNP-based heritability of methylation values in the sample of 4500 participants and a subsample of 2272 controls, respectively. The M-values (or RINT-values) after removing confounding effects were used as phenotypes in these analyses. We used only 1050,921 HapMap3 SNPs as they have been shown reliable and robust to bias in estimating SNP-based heritability and genetic correlations [64,65,66]. A genetic relationship matrix based on these SNPs was created and implemented in GREML. The heritability analyses were performed using the software GCTA [62].

#### 4.4.3. Association of Non-Genetic Factors with VTRNA2-1 Methylation

We used mixed linear regressions similar to our previous publications [20,21] to assess the association of age, sex, BMI, smoking, alcohol consumption with methylation at *VTRNA2-1.* This analysis was undertaken using the same set of 4500 participants used for the genetic analyses, as well as separately in MCCS controls only.

#### 4.4.4. Associations with Breast Cancer Risk

Cox proportional hazards survival analysis was used to test for associations between variants associated with *VTRNA2-1* methylation and breast cancer risk using all participants from the 25 multi-generational multiple-case breast cancer families (*n* = 2141). This analysis was based on the phenotype and relationships data of these 2141 participants, and the methylation and genetic data on 202 of them. Unobserved methylation and SNP data were replaced by estimated carrier probabilities using the methods presented in [1]. As the families in this study were ascertained because they each contained multiple breast cancer cases, and no adjustment for this ascertainment criterion was made, hazard ratio estimates are biased, but since the ascertainment criterion has no effect on the test statistic under the null hypothesis, the *p* values for association with breast cancer are valid. These *p* values were based on the likelihood ratio test, not the Wald test, so variances for the hazard ratios were not needed and hence were not estimated using either standard maximum likelihood or robust variance. The same models as in Joo et al. [1] were performed, with additional adjustment for genetic variant carrier probabilities at rs2346018, which was one of the strongest mQTLs in this study (see Results), and previously reported to influence *VTRNA2-1* promoter methylation [7]). Similar results were obtained after adjustment for any of the other eight variants for which carrier probabilities could be estimated (data not shown).

## Figures and Tables

**Figure 1 ijms-22-02535-f001:**
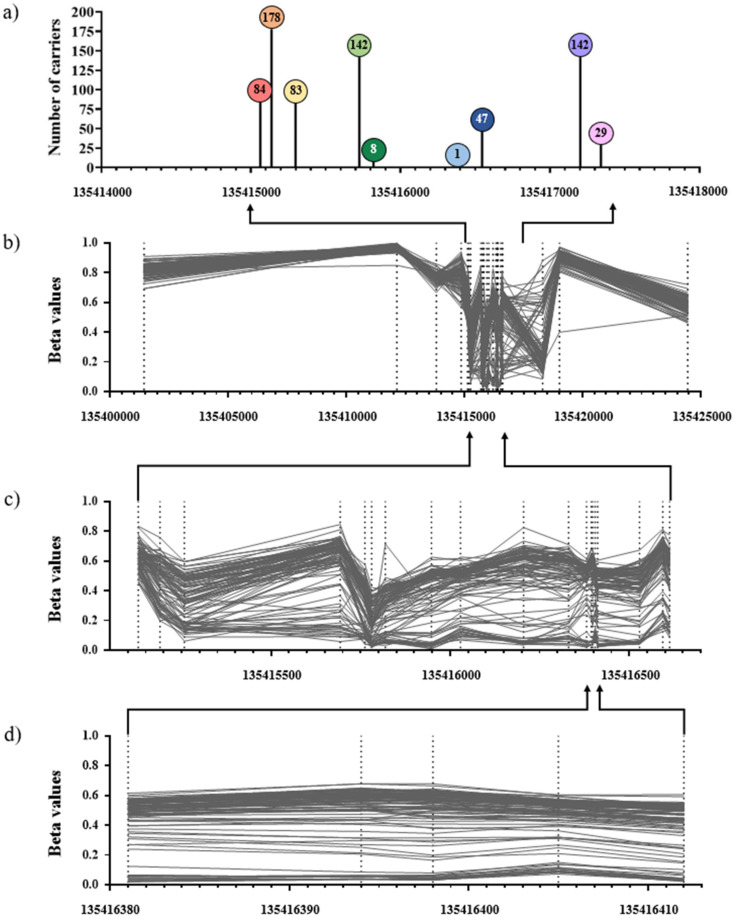
Genetic variants and DNA methylation at the *VTRNA2-1* region. Panel (**a**) shows the 9 variants identified via sequencing in the 179 participants of the multiple-case breast cancer family studies (ABCFR/kConFab). From left to right: rs62365993, rs7706795, rs2346018, rs2346019, rs34577747, rs1366231064, rs9327740, rs4976364, and rs74555710. Numbers within lollipops indicate the number of minor alleles observed in the sample. Panel (**b**)–(**d**) are from a random sample of 100 participants in the population-based study (MCCS), with coordinates: (**b**) 13401437–1354244 (26 CpGs), (**c**) 135415129–135416613 (19 CpGs), and (**d**) 135416381–135416412 (the 5 most heritable methylation marks).

**Figure 2 ijms-22-02535-f002:**
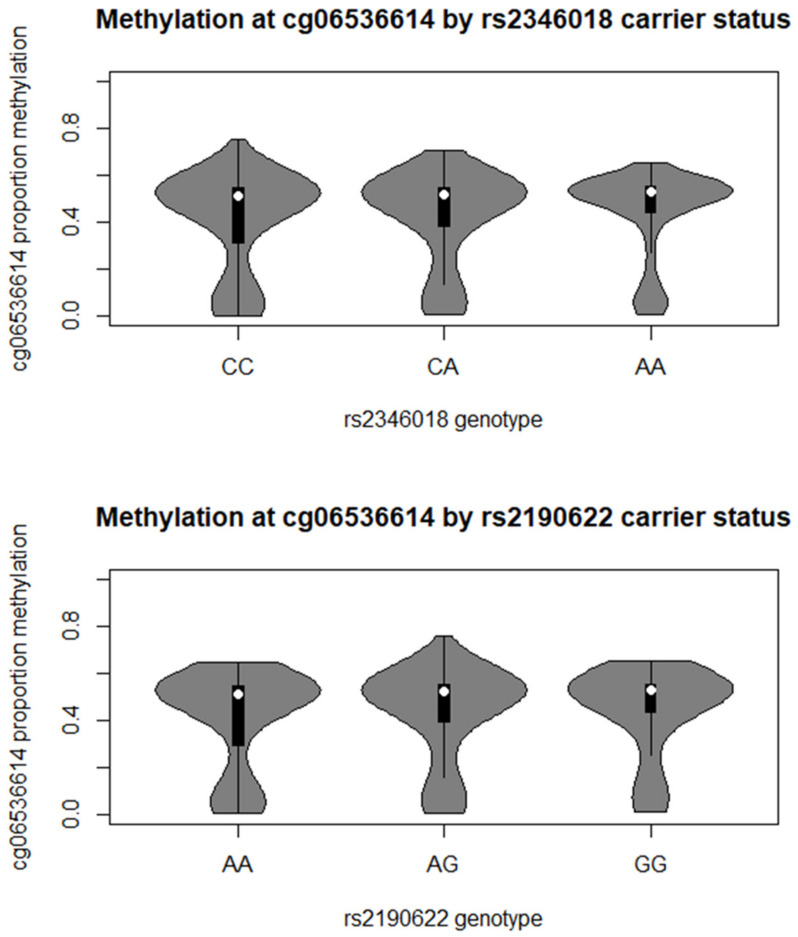
*VTRNA2-1* promoter percentage methylation distribution by carrier status at rs2346018 (CTCF binding site) and rs2190622 (strongest observed association). Carrier frequency (percentage): **rs2346018**: CC: 1969 (44%); CA: 1936 (43%); AA: 495 (11%); **rs2190622**: CC: 1838 (41%); CA: 2064 (46%); AA: 597 (13%).

**Figure 3 ijms-22-02535-f003:**
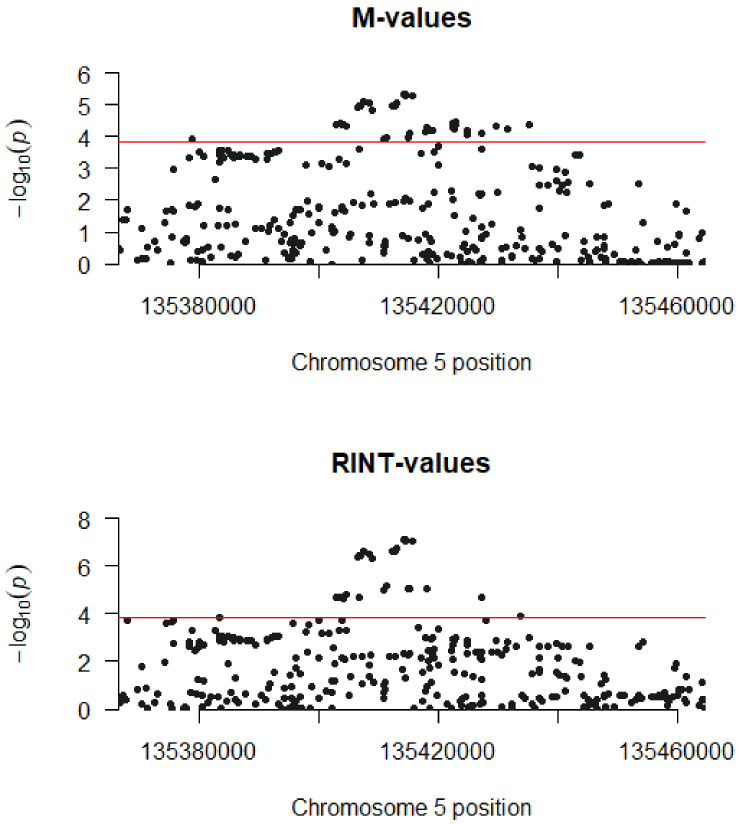
Manhattan plot for 334 genetic variants within 50 kb of cg06536614. The red line shows the Bonferroni threshold used to declare *cis-*mQTLs (*p* = 1.5 × 10^−4^); for the M-value analysis, the 46 significant associations are presented in Table 2. All variant names, positions, and quantitative results are shown in Appendix A

**Table 1 ijms-22-02535-t001:** Characteristics of the population-based and breast cancer family-based studies.

	Controls (*n* = 2272)	Cases (*n* = 2228)
**Population-based study (MCCS)**		
**Age at blood draw** (median [IQR])	59.8 [52.7–65.0]	60.3 [53.3–65.6]
**Sex** (female)	919 (40%)	901 (40%)
**Country of birth**		
Australia/NZ	1602 (70%)	1571 (69%)
UK/Northern Europe	161 (7%)	156 (7%)
Italy	311 (14%)	297 (13%)
Greece	198 (9%)	204 (9%)
**Smoking status**		
Never	1154 (51%)	1106 (49%)
Former	854 (38%)	888 (39%)
Current	263 (11%)	233 (10%)
**BMI** (kg/m2)	26.4 [24.1–29.1]	26.8 [24.4–29.7]
**Alcohol consumption** (g/day)	4.3 [0.0–17.1]	4.3 [0.0–17.1]
**Alternate Healthy Eating Index 2010**	63.5 [56.0–71.0]	64.0 [56.5–71.5]
**Family-based studies (ABCFR/kConFab) ^a^**		
**Sex** (female)	123 (100%)	87 (100%)

Abbreviations: MCCS: Melbourne Collaborative Cohort Study; ABCFR/kConFab: Australian Breast Cancer Family Registry and Kathleen Cuningham Foundation Consortium for research into Familial Breast Cancer; IQR: interquartile range; BMI: body mass index. ^a^ Of these 210 participants, 179 had sequencing data available.

**Table 2 ijms-22-02535-t002:** Strongest mQTLs within 50 kb of cg06536614 (46 variants for the analysis of M-values).

Chromosome	Position	Variant	REF ^a^	ALT ^a^	BETA ^a^	SE ^a^	*P* ^a^	R2 ^a^	MAF ^a^
5	135378781	rs3805700	A	G	0.093	0.024	1.2 × 10^-4^	0.003	0.26
5	135402852	rs11956252	G	C	0.092	0.023	4.1 × 10^-4^	0.004	0.32
5	135403529	rs6899012	G	A	0.093	0.022	3.9 × 10^-4^	0.004	0.32
5	135403745	rs74634331	AGTG	AG	0.092	0.023	4.2 × 10^-4^	0.004	0.32
5	135404173	rs9986124	G	T	0.093	0.023	4.1 × 10^-4^	0.004	0.32
5	135404613	rs9986287	T	C	0.093	0.023	4.7 × 10^-4^	0.004	0.32
5	135406459	rs7725702	C	G	0.097	0.022	1.2 × 10^-4^	0.004	0.36
5	135406534	rs7725447	G	A	0.097	0.022	1.2 × 10^-4^	0.004	0.35
5	135406658	rs2058043	A	G	0.098	0.022	1.0 × 10^-4^	0.004	0.36
5	135406894	rs2058042	G	A	0.098	0.022	1.1 × 10^-4^	0.004	0.35
5	135407572	rs4976470	A	G	0.099	0.022	8.0 × 10^-4^	0.004	0.36
5	135408325	rs4976471	T	A	0.099	0.022	8.6 × 10^-4^	0.004	0.36
5	135409014	rs6861956	T	C	0.096	0.022	1.5 × 10^-4^	0.004	0.36
5	135410863	rs11742191	A	G	0.087	0.022	1.2 × 10^-4^	0.003	0.33
5	135411281	rs11749522	C	T	0.087	0.022	1.1 × 10^-4^	0.003	0.33
5	135412195	rs10079215	A	G	0.097	0.022	1.1 × 10^-4^	0.004	0.36
5	135412675	rs35137944	A	G	0.097	0.022	1.0 × 10^-4^	0.004	0.36
5	135413026	rs7724672	A	G	0.098	0.022	8.7 × 10^-4^	0.004	0.36
5	135414280	rs2190622	A	G	0.100	0.022	4.6 × 10^-4^	0.005	0.36
5	135414455	rs4246798	A	G	0.100	0.022	4.6 × 10^-4^	0.005	0.36
5	135414510	rs4246799	G	A	0.100	0.022	5.3 × 10^-4^	0.005	0.36
5	135414866	rs17169806	C	T	0.087	0.022	1.0 × 10^-4^	0.003	0.33
**5**	**135415064**	**rs62365993**	**A**	**G**	**0.087**	**0.022**	**1.1** × 10^-4^	**0.003**	**0.33**
5	**135415300**	**rs2346018**	**C**	**A**	**0.089**	**0.023**	**8.2** × 10^-4^	**0.004**	**0.33**
5	**135415726**	**rs2346019**	**A**	**G**	**0.101**	**0.022**	**5.4** × 10^-4^	**0.005**	**0.36**
5	135417898	rs12653557	G	T	0.083	0.021	7.4 × 10^-4^	0.004	0.49
5	135418032	rs917303	G	A	0.090	0.022	5.4 × 10^-4^	0.004	0.34
5	135418717	rs4976472	G	C	0.083	0.021	6.6 × 10^-4^	0.004	0.49
5	135419159	rs4976473	C	A	0.084	0.021	6.2 × 10^-4^	0.004	0.49
5	135422443	rs11242311	T	C	0.086	0.021	3.9 × 10^-4^	0.004	0.49
5	135422507	rs34835264	G	GA	−0.087	0.022	5.3 × 10^-4^	0.004	0.50
5	135422598	rs11242312	G	A	0.086	0.021	4.2 × 10^-4^	0.004	0.49
5	135422698	rs10900843	G	A	0.084	0.021	5.7 × 10^-4^	0.004	0.49
5	135422738	rs10900844	A	G	0.086	0.021	4.2 × 10^-4^	0.004	0.49
5	135422864	rs11242313	G	A	0.087	0.021	3.4 × 10^-4^	0.004	0.49
5	135423029	rs11242314	T	C	0.086	0.021	4.4 × 10^-4^	0.004	0.49
5	135424756	rs13186426	C	A	0.083	0.021	8.5 × 10^-4^	0.004	0.48
5	135424847	5:135424847	A	AAT	0.083	0.021	7.0 × 10^-4^	0.004	0.49
5	135424922	rs1465239	A	G	0.084	0.021	6.3 × 10^-4^	0.004	0.49
5	135427371	rs1974552	T	A	0.089	0.022	8.1 × 10^-4^	0.004	0.34
5	135429640	rs1558095	C	T	0.086	0.021	4.4 × 10^-4^	0.004	0.49
5	135431590	rs1203219753	A	G	0.085	0.021	5.8 × 10^-4^	0.004	0.49
5	135435140	rs1544486	C	T	0.087	0.021	4.1 × 10^-4^5	0.004	0.49

^a^ Abbreviations: REF: Allele in the reference genome; ALT: Other allele found at that locus; BETA: Coefficient of the regression analysis of methylation on genetic variant; SE: standard error of BETA; *p*: *p*-value; R2: variance explained in methylation by genetic variant; MAF: Minor allele frequency. The three variants identified via sequencing that were available after OncoArray imputation are highlighted in bold (rs62365993, rs2346018, and rs2346019).

**Table 3 ijms-22-02535-t003:** SNP-based heritability (h^2^) for 26 CpGs in the *VTRNA2-1* (*MIR886* in the Illumina HM450 annotation file) region, for M values and RINT values.

CpG	Chromosome	Position	Name	Location	Relation to Island	Enhancer	h^2^ (M-Values)	95% CI (M-Values)	h^2^ (RINT-Values	95% CI (RINT-Values)
cg08836729	5	135401437				Yes	0	−0.14; 0.14	0	−0.14; 0.14
cg16402693	5	135412139			N_Shelf		0	−0.14; 0.14	0	−0.14; 0.14
cg17974054	5	135413810			N_Shore		0	−0.14; 0.14	0	−0.14; 0.14
cg11852404	5	135414858			N_Shore		0	−0.14; 0.14	0	−0.14; 0.14
cg16684184	5	135415129			Island		0	−0.14; 0.14	0.03	−0.11; 0.17
cg00308130	5	135415190			Island		0	−0.14; 0.14	0	−0.14; 0.14
cg15837280	5	135415258			Island		0	−0.14; 0.14	0	−0.14; 0.14
cg07158503	5	135415693			N_Shore		0	−0.14; 0.14	0	−0.14; 0.14
cg04515200	5	135415762			N_Shore		0	−0.14; 0.14	0	−0.14; 0.14
cg13581155	5	135415781			N_Shore		0	−0.14; 0.14	0	−0.14; 0.14
cg11978884	5	135415819			N_Shore		0	−0.14; 0.14	0	−0.14; 0.14
cg11608150	5	135415948			N_Shore		0	−0.14; 0.14	0	−0.14; 0.14
cg06478886	5	135416029			N_Shore		0	−0.14; 0.14	0	−0.14; 0.14
cg04481923	5	135416205	MIR886	Body	Island		0	−0.14; 0.14	0	−0.14; 0.14
cg18678645	5	135416331	MIR886	TSS200	Island		0	−0.14; 0.14	0	−0.14; 0.14
**cg06536614**	**5**	**135416381**	**MIR886**	**TSS200**	**Island**		**0**	**−0.14; 0.14**	**0**	**−0.14; 0.14**
**cg26328633**	**5**	**135416394**	**MIR886**	**TSS200**	**Island**		**0**	**−0.14; 0.14**	**0**	**−0.14; 0.14**
**cg25340688**	**5**	**135416398**	**MIR886**	**TSS200**	**Island**		**0**	**−0.14; 0.14**	**0**	**−0.14; 0.14**
**cg26896946**	**5**	**135416405**	**MIR886**	**TSS200**	**Island**		**0**	**−0.14; 0.14**	**0**	**−0.14; 0.14**
**cg00124993**	**5**	**135416412**	**MIR886**	**TSS200**	**Island**		**0**	**−0.14; 0.14**	**0**	**−0.14; 0.14**
cg08745965	5	135416529	MIR886	TSS1500	S_Shore		0	−0.14; 0.14	0	−0.14; 0.14
cg16615357	5	135416594	MIR886	TSS1500	S_Shore		0	−0.14; 0.14	0	−0.14; 0.14
cg18797653	5	135416613	MIR886	TSS1500	S_Shore		0	−0.14; 0.14	0	−0.14; 0.14
cg12897067	5	135418308			S_Shore		0.95	0.81; 1.09	0.76	0.62; 0.90
cg05631625	5	135419019			S_Shelf		0.14	0.00; 0.28	0.15	0.01; 0.29
cg01930756	5	135424444				Yes	0.09	−0.05; 0.23	0.1	−0.04; 0.24

The 5 CpGs in bold are those found to be most strongly heritable in our previous study Joo et al., Nat Commun, 2018 [1]. Name, Location, Relation to island, and Enhancer status are those provided by the Illumina HM450 annotation file.

**Table 4 ijms-22-02535-t004:** Association of non-genetic factors with *VTRNA2-1* blood DNA methylation (cg06536614) in the prospective, population-based study; 4500 participants in the Melbourne Collaborative Cohort Study (MCCS).

	Estimate ^a^	95% CI	*p*-Value
**Age (years)**	−0.005	−0.012; 0.002	0.18
**Sex (female)**	0.047	−0.114; 0.207	0.57
**Greece vs. Aus/NZ**	−0.043	−0.250; 0.164	0.68
**Italy vs. Aus/NZ**	−0.068	−0.239; 0.102	0.43
**Northern Europe vs. Aus/NZ**	0.171	−0.049; 0.391	0.13
**Current vs. never smoker**	−0.031	−0.223; 0.162	0.76
**Former vs. never smoker**	−0.050	−0.176; 0.076	0.43
**BMI (in kg/m2)**	0.002	−0.012; 0.016	0.79
**Alcohol consumption (g/day)**	0.001	−0.003; 0.004	0.70
**Healthy eating index**	0.003	−0.003; 0.008	0.36
**CD4 + T cells**	−1.760	−6.108; 2.589	0.43
**CD8+ T cells**	−0.380	−4.054; 3.294	0.84
**NK cells**	−0.858	−5.242; 3.525	0.70
**B cells**	−2.616	−6.391; 1.159	0.17
**Granulocytes**	−1.852	−5.969; 2.264	0.38
**Monocytes**	−0.355	−4.699; 3.990	0.87

^a^ Mixed linear regression model with methylation M-values as the outcome and mutually adjusted covariates modelled as fixed effects and technical variables study, assay plate and chip modelled as random effects.

**Table 5 ijms-22-02535-t005:** Associations with breast cancer risk after adjustment for rs2346018 carrier probabilities ^a^.

CpG	Chromosome	Position	∆l ^a^	Not Adjusted for SNPs ^b^		Adjusted for rs2346018	
				Biased HR (95% CI) ^c^	*p*-Value	Biased HR (95% CI) ^c^	*p*-Value
cg06536614	5	135416381	143.6	3.1 (2.1–4.6)	7 × 10^−9^	3.0 (2.0–4.3)	3 × 10^−8^
cg00124993	5	135416412	108.0	3.2 (2.2–4.7)	2 × 10^−8^	3.0 (2.0–4.4)	9 × 10^−8^
cg26328633	5	135416394	107.5	3.2 (2.2–4.8)	2 × 10^−8^	3.0 (2.0–4.5)	4 × 10^−8^
cg25340688	5	135416398	105.9	3.2 (2.1–4.7)	3 × 10^−8^	2.9 (2.0–4.3)	1 × 10^−7^
cg26896946	5	135416405	92.1	3.6 (2.4–5.4)	2 × 10^−9^	3.3 (2.2–5.0)	8 × 10^−9^

^a^ ∆l: heritability metric: details of the methods used to calculate the heritability metric, carrier probabilities and Cox models for association with breast cancer risk are provided in [1]. ^b^ These are published and unpublished results from our study Joo et al., 2018 [1] and are presented here for comparison with the results adjusted for rs2346018. ^c^ While *p* values are unbiased, hazard ratios are biased by the ascertainment of families for this study [1], and the HR estimates are only included here to show that they are virtually unchanged by adjustment for rs2346018.

## Data Availability

Data will be made available upon reasonable request to the corresponding author.

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
