# Peer review of "VTRNA2-1: Genetic Variation, Heritable Methylation and Disease Association"

_ijms, 2021, doi:10.3390/ijms22052535_

Round 1

Reviewer 1 Report

Thanks to the authors for responding to my comments. I liked the paper before and now like it even more with the improvements which address my major concerns about making the overall study design and purpose a bit clearer.

I am still a bit perplexed by Table 5 where the authors report HRs in a footnote but not in the main table. The authors comment that they believe the numbers would be important for the reader to see how little the HRs are attenuated. I agree 100% so why not put them in the table, and importantly, with CIs (as indicated in the table header) to demonstrate no attenuation? The authors mentioned that they could put these in a Supplementary Table, but I can't see why - this is a key part of the paper after all?

Anyway it's up to the editor and authors to decide. Either way I think the article is ready to be published.

Author Response

We have now included the biased HRs (95% CI) in the table and removed them from the footnote.  The footnotes have been edited to suit this new version.

Reviewer 2 Report

I wish to thank the authors for addressing my concerns and for making the suggested improvements.

Author Response

Thank you.

This manuscript is a resubmission of an earlier submission. The following is a list of the peer review reports and author responses from that submission.

Round 1

Reviewer 1 Report

Manuscript Review:

Dugué et al, VTRNA2-1: genetic variation, heritable methylation and disease association

Summary

This study analysed the potential influence of age, adult lifestyle and genetic variation on DNA methylation (DNAm) at the VTRNA2-1 locus, a region where DNAm has previously been associated with multiple health outcomes including cancers, and with nutritional and other exposures. The authors found no evidence that VTRNA2-1 DNAm was associated with age or any of the adult lifestyle considered, but did identify multiple cis-mQTL with small effects.

The authors previously found evidence that methylation was heritable at several CpGs in this locus in a family-based analysis, which makes their finding of no evidence for SNP-based heritability / strong mQTL effects particularly interesting, all the more so in light of potential links to heritable breast cancer.

There is great interest in links between DNAm, prenatal and lifelong exposures and disease at this locus, and in elucidating mechanisms that may influence establishment of parent-of-origin methylation states. This study is therefore timely and provides interesting further insights into these issues.

(Note numbers below refer to line numbers in the manuscript).

Major comments

All my major comments relate to a slight lack of clarity on the rationale for the overall study design which makes the manuscript harder to read than it should be.

The aims outlined in concluding para of Intro (90-95) don’t align very well with those stated in the abstract. This is related to points made below.

The rationale for the sequencing of individuals in the family study is not very clear, even though this occupies a prominent position in the manuscript (20-21,100-107, Fig. 1a, 171-4, Methods). As I understand it these variants were identified to check that they were present in the dataset used in the mQTL analysis (105-6), i.e. relevant to assessing the degree to which mQTLs might contribute to familial/heritable risk? Furthermore, only a single SNP from the sequenced genotype data (rs2346018) was used in the adjusted breast cancer risk analysis, presumably because this is the only sequenced variant present in the family cohorts that was also a cis-mQTL? Either way I think the rationale linking these different elements should be made clearer throughout the text as it’s currently rather confusing.

Related to the above, it should be made clearer how the breast cancer risk analysis (169-182, Table 5) differs from the previously reported analysis (Joo et al Nat Comms), and the degree to which previous findings are replicated. Again as I understand it the key purpose is to understand the degree to which DNAm-associated breast cancer risk might be influenced by the mQTL. It is important to make these points clear in the text, otherwise the reporting of the link between DNAm and breast cancer risk at this CpG is hard to justify given that it has already been published.

218-219, 245-254 Given the author’s central interest in methylation heritability and breast cancer, this element of the discussion seems a little underplayed. For example, I think the issue of methylation ‘heritability’ could do with some further explanation, and possibly also in the opening para of the introduction. As I understand it the authors wish to make a key distinction between SNP-based heritability (mQTLs) and ‘epimutations’ (as described in Joo et al). The authors and others have shown evidence that mQTL effects are minimal which is an interesting finding in combination with the heritability estimates at this locus from Joo et al, but this point seems to be a bit lost, along with possible implications for breast cancer heritability given reported associations here and in Joo et al. I would like to see a clearer exposition of these ideas in the discussion.

Minor comments

97-98 It would help readability to align the study types / names on first mention, i.e. prospective cohort (MCCS) and multiple-case family studies (ABCFR/kConFab). It might also help thereafter to stick to one type of descriptor, e.g. MCCS/ABCFR/kConFab, or perhaps MCCS/FS (for family study) or similar.

Fig. 1

  1. a) specify what the sample is in the caption (I think all ABCFR/kConFab from the text). Also define ‘carriers’. Individuals with one and/or two variant alleles?
  2. d) caption suggests 5 most heritable marks are plotted in c) but surely this is d)?

120-123 This would appear to be referring to Fig 1d but text says Figs 1 and 2. Also it would help to label the CpG in question in the figure otherwise there seems little point in referring to the figure.

128 Brief mention should be made of variants and samples analysed for the gw mQTL analysis in the main text (i.e. number of variants, cohort and sample size). This information is available in Methods but it’s not easy to find and it would be helpful to have it in concise form in the main text.

129-130 The authors are to be commended for performing parallel analyses with M-value and RINT transformations. However these terms should be explained on first mention, and their use justified with references.

130 The text suggests that results from the gw mQTL analysis are presented in Supplementary Tables 1 and 2 (130) whereas these tables each include 100 SNPs. It looks like these are the top 100 by p-value. If so this should be mentioned somewhere.

Figure 2 As mentioned above, the current reference to this figure in the main text appears to be wrong. It seems like this figure should instead be referred to when describing the minimal effect of genetic variation at this locus – i.e. somewhere in 128-144

Figure 3 It would be helpful to mark the position of the CpG considered in the analysis.

Table 3 caption is too brief. Further information required includes:

  • Why two sets of h^2 and CIs?
  • Reason for CpGs highlighted in bold
  • Source / meaning for Name, Location, Relation to island, Enhancer (looks like 450k manifest?)

153 h^2 / CI refers to median/modal values across the region. While the overall conclusion (minimal heritability) is well supported, this should be changed and some reference made to the high heritability at one CpG at one end of the region analysed.

201-2 Arguably it is not quite correct to say that rs2190622 has unknown regulatory function. The authors already point out that this SNP has been linked to polymorphic imprinting via an effect on CTCF binding.

Table 4 Sample cohort and numbers should be given in the caption. May be a journal formatting issue, but information in the footnote would be better in the main caption following the title as it applies to the whole table.

It seems odd to include Table 5 with HRs missing, but included as a footnote. The authors should decide whether or not the HR results are worth including in the main table (with suitable caveats), or if not, move them to supplementary info.

249 The authors will be aware that the distinction between trans- and intergenerational inheritance is important, with the former a highly controversial topic. I suggest ‘intergenerational’ is more appropriate here (and they have used it elsewhere – 272)

255-7 ‘blood samples, i.e. germline’ – it is not clear what point is being made here since changes in DNAm could could occur in healthy individuals.

What is the purpose of highlighted loci in Supplementary Tables 3 and 4?

Author Response

All my major comments relate to a slight lack of clarity on the rationale for the overall study design which makes the manuscript harder to read than it should be.

The aims outlined in concluding para of Intro (90-95) don’t align very well with those stated in the abstract. This is related to points made below.

The rationale for the sequencing of individuals in the family study is not very clear, even though this occupies a prominent position in the manuscript (20-21,100-107, Fig. 1a, 171-4, Methods). As I understand it these variants were identified to check that they were present in the dataset used in the mQTL analysis (105-6), i.e. relevant to assessing the degree to which mQTLs might contribute to familial/heritable risk? Furthermore, only a single SNP from the sequenced genotype data (rs2346018) was used in the adjusted breast cancer risk analysis, presumably because this is the only sequenced variant present in the family cohorts that was also a cis-mQTL? Either way I think the rationale linking these different elements should be made clearer throughout the text as it’s currently rather confusing.

The reviewer is correct that sequencing of individuals in the family study was done to identify the variants present in this region and check that these were also included in the mQTL analysis (using the OncoArray; prospective, population-based study with a large sample size of 4,500 participants). This was the case as 8 of the 9 variants identified via sequencing were also available after genotype imputation; the variant that was not retrieved using the OncoArray was rare (1 carrier in 179 sequenced participants).

This is reflected in the Introduction: “In this study, our aims were three-fold: first, to sequence the VTRNA2-1 region to assess the presence of rare genetic variation at this locus; second, to conduct a genome-wide assessment of mQTLs and SNP-based methylation heritability in the VTRNA2-1 region (previously identified heritable marks); third, to assess whether any genetic variants associated with DNA methylation in this region contribute to the previously observed associations with breast cancer risk.”

In the Results: “Nine genetic variants were identified via sequencing (Figure 1a) in members of multiple-case breast cancer families. Of these, one was rare (identified in only one participant) and was excluded from further analysis. The other eight variants were used to estimate carrier probabilities in members of multiple-case breast cancer families (see Methods; Associations with breast cancer risk). All eight variants were available after genotype imputation in the prospective cohort study (see Methods; Genetic and methylation data), and therefore included in the mQTL analysis.”, and

“Associations of VTRNA2-1 methylation (cg06536614) with breast cancer risk were assessed in 2,141 participants in multiple-case breast cancer families, using the same methods as in our previous publication [1]. As genotypes were not directly measured in all family members, we estimated carrier probabilities for genetic variant, using a method similar to that used to estimate methylation carrier probabilities [1], based on sequencing data for 179 participants and imputed OncoArray data for 23 participants (all multiple-case family members); the association of methylation with breast cancer risk was therefore assessed using the same models as previously, with additional adjustment for genetic variant carrier probabilities. The association of cg06536614 methylation with breast cancer risk remained highly significant after adjustment for rs2346018 carrier probabilities, with P values ranging from 2x10-9 to 3x10-8 in the unadjusted model, and from 8x10-9 to 1x10-7 in the adjusted model (Table 5). Similar results were obtained after adjustment for any of the other three variants significantly associated with cg06536614 methylation (Table 2) for which carrier probabilities could be estimated (data not shown).”

We have added as a footnote to Table 2: “The three variants identified via sequencing that were available after OncoArray imputation are highlighted in bold (rs62365993, rs2346018, and rs2346019).”

We have also added text to clarify this at the start of Results: “This study used data from: i) 179 participants of multiple-case breast cancer family studies to assess the presence of genetic variants in the VTRNA2-1 heritable region, ii) 4,500 participants in a prospective study to assess mQTLs and SNP-based heritability (genome-wide and focusing on cis-variants); and iii) 2,141 participants in breast cancer family-based studies to adjust the VTRNA2-1 results of our previous publication [1] for a nearby SNP.”

And in the Methods: “This analysis was based on the phenotype and relationships data of these 2141 participants, and the methylation and genetic data on 202 of them.”

Related to the above, it should be made clearer how the breast cancer risk analysis (169-182, Table 5) differs from the previously reported analysis (Joo et al Nat Comms), and the degree to which previous findings are replicated. Again as I understand it the key purpose is to understand the degree to which DNAm-associated breast cancer risk might be influenced by the mQTL. It is important to make these points clear in the text, otherwise the reporting of the link between DNAm and breast cancer risk at this CpG is hard to justify given that it has already been published.

The rationale of the analyses of association with breast cancer risk was to assess whether our previously observed associations remained the same after adjustment for genetic variants. Therefore the results presented in the first column of Table 5 (“Not adjusted for SNPs”) are the same as those we reported previously (Joo et al. Nat Commun. 2018). We presented results after adjustment for rs2346018 because this SNP was one of the top associations in the cis-mQTL analysis and was previously demonstrated to have functional relevance at this locus (Carpenter, PNAS, 2018). Results after adjustment for other SNPs were similar.

We have clarified this: By adding a footnote to Table 5: Not adjusted for SNPsb

b These results are the same as those presented in Joo et al. Nat Commun. 2018 [1] and presented here for comparison with adjusted associations.

In the Results: “The results ‘Not adjusted for SNPs’ are the same as those presented in Joo et al. [1].”

218-219, 245-254 Given the author’s central interest in methylation heritability and breast cancer, this element of the discussion seems a little underplayed. For example, I think the issue of methylation ‘heritability’ could do with some further explanation, and possibly also in the opening para of the introduction. As I understand it the authors wish to make a key distinction between SNP-based heritability (mQTLs) and ‘epimutations’ (as described in Joo et al). The authors and others have shown evidence that mQTL effects are minimal which is an interesting finding in combination with the heritability estimates at this locus from Joo et al, but this point seems to be a bit lost, along with possible implications for breast cancer heritability given reported associations here and in Joo et al. I would like to see a clearer exposition of these ideas in the discussion.

We have clarified this at the start of Introduction: “Mendelian-like inheritance of germline DNA methylation can be due to cis- or trans-acting genetic factors known as methylation Quantitative Trait Loci (mQTL) or epimutations (heritable change in gene activity that is not associated with a DNA mutation but rather with gain or loss of DNA methylation or other heritable modification of chromatin). Both can mimic germline pathogenic variants in their effect on gene function and disease association and discriminating between the two possibilities (mQTL or epimutation) in specific genomic regions and disease context is often challenging.”

And at the start of the Discussion: “Our study provides further evidence that DNA methylation at VTRNA2-1 is minimally influenced by genetic factors and thus the Mendelian-like inheritance of germline DNA methylation at this loci is likely to be via a true epimutation mechanism rather than via a mQTL. Therefore, the ‘missing heritability’ (approximately the difference between family-based and SNP-based heritability) appears to be substantial, which confirms the findings of Joo et al. [1]. Genetic variants are therefore unlikely to fully explain any associations between methylation at this locus and disease risk, including breast cancer as we found previously in the context of a multiple-case breast cancer family study.”

Minor comments

97-98 It would help readability to align the study types / names on first mention, i.e. prospective cohort (MCCS) and multiple-case family studies (ABCFR/kConFab). It might also help thereafter to stick to one type of descriptor, e.g. MCCS/ABCFR/kConFab, or perhaps MCCS/FS (for family study) or similar.

We have clarified this line 97-98 and in Table 1.

Fig. 1

  1. a) specify what the sample is in the caption (I think all ABCFR/kConFab from the text). Also define ‘carriers’. Individuals with one and/or two variant alleles?

We have clarified the caption as follows: “Figure 1. Genetic variants and DNA methylation at the VTRNA2-1region. Panel a) shows the 9 variants identified via sequencing in the 179 participants of the multiple-case breast cancer family studies (ABCFR/kConFab). From left to right: rs62365993, rs7706795, rs2346018, rs2346019, rs34577747, rs1366231064, rs9327740, rs4976364, and rs74555710. Numbers within lollipops indicate the number of minor alleles observed in the sample. Panel b), c) and d) are from a random sample of 100 participants in the population-based study (MCCS), with coordinates: b) 13401437-1354244 (26 CpGs), c) 135415129-135416613 (19 CpGs), and d) 135416381-135416412 (the 5 most heritable methylation marks).”

  1. d) caption suggests 5 most heritable marks are plotted in c) but surely this is d)?

Yes this is correct. This was inaccurately changed by the production team and we have corrected it back.

 120-123 This would appear to be referring to Fig 1d but text says Figs 1 and 2. Also it would help to label the CpG in question in the figure otherwise there seems little point in referring to the figure.

Figure 1 (b-c) shows that methylation profiles at cg06536614, cg26328633, cg25340688, cg26896946, cg00124993 are highly similar.

Figure 1 (b-c) and 2 allow the reader to see the distribution of methylation values at these five CpGs (Figure 1 (b-c)) and for the most heritable site (cg06536614; Figure 2); formal quantification of the proportions of participants between specified methylation thresholds is provided in the text. We have removed mention of Figure 1 lines 126-127 as these proportions are only provided for cg06536614.

128 Brief mention should be made of variants and samples analysed for the gw mQTL analysis in the main text (i.e. number of variants, cohort and sample size). This information is available in Methods but it’s not easy to find and it would be helpful to have it in concise form in the main text.

We have clarified this by adding an introducing sentence in this section of the Results as follows:

“The mQTL analysis was performed in 4,500 participants in the prospective, population-based study (MCCS) for a total of 10,484,498 genetic variants; a cis-mQTL analysis was then carried out by focusing on genetic variants within 50kb of the most heritable methylation mark (cg06536614).”

129-130 The authors are to be commended for performing parallel analyses with M-value and RINT transformations. However these terms should be explained on first mention, and their use justified with references.

We have corrected this as follows:

“Genome-wide, we found no evidence that any included genetic variant was associated with VTRNA2-1 methylation (all P>5x10-9), with similar results obtained for the M-value (logit transformation of beta value) or RINT (rank-based inverse normal transformation, which was applied previously in the context of DNA methylation analyses [55, 56] and provides a Gaussian methylation distribution, which is not always the case for the M-values) transformation…”

130 The text suggests that results from the gw mQTL analysis are presented in Supplementary Tables 1 and 2 (130) whereas these tables each include 100 SNPs. It looks like these are the top 100 by p-value. If so this should be mentioned somewhere.

We have clarified this: “(Supplementary Table 1 and 2, respectively, showing the 100 CpGs with smallest P-values)”

Figure 2 As mentioned above, the current reference to this figure in the main text appears to be wrong. It seems like this figure should instead be referred to when describing the minimal effect of genetic variation at this locus – i.e. somewhere in 128-144

Figure 2 is useful for the reader to 1) have a feel for the methylation distribution at this site (correctly referred to lines 126-127), and 2) to see that there is minimal influence of genetic variation, so we have referred to this Figure 2 on line 183, line 184, and line 190.

Figure 3 It would be helpful to mark the position of the CpG considered in the analysis.

We have added to the caption of Figure 3 the following clarification:

“Figure 3. Manhattan plot for 334 genetic variants within 50kb of cg06536614. The red line shows the Bonferroni threshold used to declare cis-mQTLs (P=1.5x10-4); for the M-value analysis, the 46 significant associations are presented in Table 2. All variant names, positions, and quantitative results are shown in Supplementary Table 3 and 4.”

Table 3 caption is too brief. Further information required includes:

Why two sets of h^2 and CIs?

We have clarified this by adding “M-values” or “RINT values” in the Table header; this was provided in our original submission but was cut by the production team. We have also clarified the table title.

Reason for CpGs highlighted in bold

Now clarified as follows: “The 5 CpGs in bold are those found to be most strongly heritable in our previous study Joo et al., Nat Communs, 2018 [1].”

Source / meaning for Name, Location, Relation to island, Enhancer (looks like 450k manifest?).

Now clarified as follows: “Name, Location, Relation to island, and Enhancer status are those provided by the Illumina HM450 annotation file.”

153 h^2 / CI refers to median/modal values across the region. While the overall conclusion (minimal heritability) is well supported, this should be changed and some reference made to the high heritability at one CpG at one end of the region analysed.

h2 and 95% CI refer to the point estimate and confidence interval for each CpG, not the overall region. The SNP-based heritability analysis is performed by GCTA using 1,050,921 HapMap3 SNPs that have been shown reliable and robust to bias in estimating SNP-based heritability and genetic correlations.

We included CpGs slightly outside of the region in Table 3, but those with non-null heritability are far from the most heritable methylation mark (2 to 8 kb away from cluster) and are not annotated to the VTRNA2-1 gene, which is why we do not draw conclusions from these. We have added in Results: “Non-null or high heritability was observed for methylation sites distant from the heritable VTRNA2-1 region (>2-8kb, Table 3)”.

201-2 Arguably it is not quite correct to say that rs2190622 has unknown regulatory function. The authors already point out that this SNP has been linked to polymorphic imprinting via an effect on CTCF binding.

We believe this point was correctly explained in the manuscript. rs2346018 (rather than rs2190622) has been linked to polymorphic imprinting via an effect on CTCF (lines 88-89) and previously reported to influence VTRNA2-1 promoter methylation (Lines 483-486). See also Supplementary Table 5.

Table 4 Sample cohort and numbers should be given in the caption. May be a journal formatting issue, but information in the footnote would be better in the main caption following the title as it applies to the whole table.

We have clarified the caption of Table 4: “Association of non-genetic factors with VTRNA2-1 blood DNA methylation (cg06536614) in the prospective, population-based study; 4,500 participants in the Melbourne Collaborative Cohort Study (MCCS).”

It seems odd to include Table 5 with HRs missing, but included as a footnote. The authors should decide whether or not the HR results are worth including in the main table (with suitable caveats), or if not, move them to supplementary info.

Although this choice was somewhat unorthodox, we believe the numbers would be important for the reader to see how little the HRs are attenuated. We would prefer to keep as a footnote but would have no issue adding the same table including HRs in a Supplementary file if the Editor requested it.

249 The authors will be aware that the distinction between trans- and intergenerational inheritance is important, with the former a highly controversial topic. I suggest ‘intergenerational’ is more appropriate here (and they have used it elsewhere – 272)

We agree with the Reviewer and have kept the question of the mechanism by which methylation is inherited open. Last sentence of Discussion: “The mechanism of inheritance of DNA methylation in this region remains to be elucidated.”

255-7 ‘blood samples, i.e. germline’ – it is not clear what point is being made here since changes in DNAm could could occur in healthy individuals.

We have removed germline (2 instances) and replaced by “blood” in the second instance. We wanted to make very clear that although our manuscript included some cancer data, DNA methylation was not measured in tumours.

What is the purpose of highlighted loci in Supplementary Tables 3 and 4?

We have removed highlighting in Supplementary Tables 3 and 4.

Reviewer 2 Report

In this manuscript, the authors investigated the link between genetic polymorphisms and methylation at VTRNA2-1. They conclude the absence of a direct relation. In my opinion, the manuscript could be better presented and doesn't increase or add to our current knowledge. 

Some comments to improve the presentations:

Figure legends need extensions and better organization. Labeling and explanations are very poor. Most of the time the link to the text is not valid. It appears to me that the text was not written by the scientists who performed the study.

For example, in Fig 3 what the authors called the Manhattan plot (!), you could label some of the SNPs that are in discussion in the manuscript.

Polymorphisms of direct overlapping with CpGs? did the authors exclude such cases from their analysis? and their effect on methylation?

a genetic map showing the relative positions of the CpGs at investigations and the rs numbers would make it easier for general readers to follow.

Author Response

Some comments to improve the presentations:

Figure legends need extensions and better organization. Labeling and explanations are very poor. Most of the time the link to the text is not valid. It appears to me that the text was not written by the scientists who performed the study.

The text was written by the scientists who performed the study, and we have clarified Figures where we could. We have also double-checked that references of Figures in the text were correct.

For example, in Fig 3 what the authors called the Manhattan plot (!), you could label some of the SNPs that are in discussion in the manuscript.

Figure 3 is absolutely a Manhattan plot, and it is not possible to show the SNP labels for all 46 cis-mQTLs (i.e. Bonferroni significant, corrected for 334 tested). The information on SNPs is also available to the reader both in Table 2 (that comes just before Figure 3) and in the Supplementary Material.

Polymorphisms of direct overlapping with CpGs? did the authors exclude such cases from their analysis? and their effect on methylation?

There were no variants directly overlapping with the most heritable VTRNA2-1 methylation site (cg06536614), which was assessed for mQTLs in our study.

We have added in Results: “None of these variants were found to directly overlap with the most heritable VTRNA2-1methylation site (cg06536614).”

a genetic map showing the relative positions of the CpGs at investigations and the rs numbers would make it easier for general readers to follow.

This information is already provided in the manuscript. For example the relative position of CpG and rs numbers are in Table 2, Figure 1 and Figure 3 (pages 4, 6 and 7).

Reviewer 3 Report

The authors had three aims: 1) to sequence the VTRNA2-1 region to assess the presence of rare genetic variation at this locus;2)  to conduct a genome-wide assessment of mQTLs and SNP-based methylation heritability in the VTRNA2-1 region (previously identified heritable marks); 3) to assess whether any genetic variants associated with DNA methylation in this region contribute to the previously observed associations with breast cancer risk. The conclusion was that  genetic factors (and adult lifestyle) play a minimal role in explaining methylation variability at the heritable VTRNA2-1 cluster.

The paper is of high interest to the field and the study is well-designed. However, I have some suggestions for improving the presentation:

1) As for the data in Table 1, it is not clear if all subjects are from the breast cancer families.

2) Were any female only analyses attempted, even that sex was not found to affect DNA methylation? Or how about doing analyses in cases only (as was done for controls only). If not, an explanation why these were not done, would be appreciated.

3) For Figure 1, could the SNPs rs-numbers be shown? Also provide an explanation for the numbers at the SNP sites, in the figure legend. 

4) Overall, it is hard to follow which subjects were included into each analysis. Perhaps a flow chart could be provided to show participants in different analyses; also the selected variables for each analysis could be included on such a flow chart.

5) Lastly, how were the HM450 probes filtered if any? If probes with SNPs were excluded, how could this affect the mQTL analysis?

Author Response

The authors had three aims: 1) to sequence the VTRNA2-1 region to assess the presence of rare genetic variation at this locus;2)  to conduct a genome-wide assessment of mQTLs and SNP-based methylation heritability in the VTRNA2-1 region (previously identified heritable marks); 3) to assess whether any genetic variants associated with DNA methylation in this region contribute to the previously observed associations with breast cancer risk. The conclusion was that genetic factors (and adult lifestyle) play a minimal role in explaining methylation variability at the heritable VTRNA2-1 cluster.

The paper is of high interest to the field and the study is well-designed. However, I have some suggestions for improving the presentation:

1) As for the data in Table 1, it is not clear if all subjects are from the breast cancer families.

In Table 1, the upper panel is for a population-based study (Melbourne Collaborative Cohort Study) and the lower panel is for breast cancer family-based studies (Australian Breast Cancer Family Registry & Kathleen Cuningham Foundation Consortium for research into Familial Breast Cancer).

We have clarified this by modifying the left column and Table title.

2) Were any female only analyses attempted, even that sex was not found to affect DNA methylation? Or how about doing analyses in cases only (as was done for controls only). If not, an explanation why these were not done, would be appreciated.

We did not do women-only analyses because we considered that if genetic variants or age and lifestyle factors were associated with blood DNA methylation, this should be true in both males and females (hence, any observed association would be a false-positive)

We did not do case-only analyses because these would potentially be affected by collider bias. The analysis was performed in controls only (no collider bias) and for the whole dataset (to maximise sample size, being twice greater by including both controls and cases; with no strong evidence of collider bias in cases in this context).

3) For Figure 1, could the SNPs rs-numbers be shown? Also provide an explanation for the numbers at the SNP sites, in the figure legend.

We have clarified these points in the footnote of Figure 1 as follows:

“Panel a) shows the 9 variants identified via sequencing in the 179 participants of the multiple-case breast cancer family studies (ABCFR/kConFab). From left to right: rs62365993, rs7706795, rs2346018, rs2346019, rs34577747, rs1366231064, rs9327740, rs4976364, and rs74555710. Numbers within lollipops indicate the number of minor alleles observed in the sample. Panel b), c) and d) are from a random sample of 100 participants in the population-based study (MCCS), with coordinates: b) 13401437-1354244 (26 CpGs), c) 135415129-135416613 (19 CpGs), and d) 135416381-135416412 (the 5 most heritable methylation marks).”

4) Overall, it is hard to follow which subjects were included into each analysis. Perhaps a flow chart could be provided to show participants in different analyses; also the selected variables for each analysis could be included on such a flow chart.

Sequencing in breast cancer family members (ABCFR/kConFab): 179 participants

mQTL analysis: 4500 participants in the prospective study (MCCS)

Association with breast cancer: 2141 participants, based on phenotypes (breast cancer affected status and ages) and relationship data on all 2141 participants and genetic and methylation data on 202 of these participants.

We have added text to clarify this at the start of Results: “This study used data from: i) 179 participants in breast cancer family-based studies to assess the presence of genetic variants in the VTRNA2-1 heritable region, ii) 4,500 participants in a prospective study to assess mQTLs and SNP-based heritability (genome-wide and focusing on cis-variants); and iii) 2,141 participants in breast cancer family-based studies to adjust the VTRNA2-1 results of our previous publication [1] for a nearby SNP.”

And in the Methods: “This analysis was based on the phenotype and relationships data of these 2141 participants, and the methylation and genetic data on 202 of them.”

5) Lastly, how were the HM450 probes filtered if any? If probes with SNPs were excluded, how could this affect the mQTL analysis?

Our study focused on VTRNA2-1 and all CpGs available on the HM450 assay in the region were included, i.e. 26 CpGs in the ‘extended region’; 10 CpGs annotated by Illumina to VTRNA2-1 (MIR886) and mQTL analysis focused on the CpG found to be most strongly heritable in Joo et al. Nat Commun, 2018, see Table 3. No SNPs overlapped with these CpGs.